# Foam-in-Vein: Characterisation of Blood Displacement Efficacy of Liquid Sclerosing Foams

**DOI:** 10.3390/biom12121725

**Published:** 2022-11-22

**Authors:** Alireza Meghdadi, Stephen A. Jones, Venisha A. Patel, Andrew L. Lewis, Timothy M. Millar, Dario Carugo

**Affiliations:** 1Faculty of Engineering and Physical Sciences, University of Southampton, Southampton SO17 1BJ, UK; 2Department of Pharmaceutics, School of Pharmacy, University College London, London WC1N 1AX, UK; 3Biocompatibles UK Ltd., Lakeview, Riverside Way, Watchmoor Park, Camberley GU15 3YL, UK; 4Faculty of Medicine, University of Southampton, Southampton SO17 1BJ, UK

**Keywords:** sclerotherapy, foam, rheology, displacement flow, varicose vein, physician compounded foam

## Abstract

Sclerotherapy is among the least invasive and most commonly utilised treatment options for varicose veins. Nonetheless, it does not cure varicosities permanently and recurrence rates are of up to 64%. Although sclerosing foams have been extensively characterised with respect to their bench-top properties, such as bubble size distribution and half-life, little is known about their flow behaviour within the venous environment during treatment. Additionally, current methods of foam characterisation do not recapitulate the end-point administration conditions, hindering optimisation of therapeutic efficacy. Here, a therapeutically relevant apparatus has been used to obtain a clinically relevant rheological model of sclerosing foams. This model was then correlated with a therapeutically applicable parameter—i.e., the capability of foams to displace blood within a vein. A pipe viscometry apparatus was employed to obtain a rheological model of 1% polidocanol foams across shear rates of 6 s^−1^ to 400 s^−1^. Two different foam formulation techniques (double syringe system and Tessari) and three liquid-to-gas ratios (1:3, 1:4 and 1:5) were investigated. A power-law model was employed on the rheological data to obtain the apparent viscosity of foams. In a separate experiment, a finite volume of foam was injected into a PTFE tube to displace a blood surrogate solution (0.2% *w*/*v* carboxymethyl cellulose). The displaced blood surrogate was collected, weighed, and correlated with foam’s apparent viscosity. Results showed a decreasing displacement efficacy with foam dryness and injection flowrate. Furthermore, an asymptotic model was formulated that may be used to predict the extent of blood displacement for a given foam formulation and volume. The developed model could guide clinicians in their selection of a foam formulation that exhibits the greatest blood displacement efficacy.

## 1. Introduction

Varicose veins of the lower limbs are common venous insufficiencies with global prevalence estimates ranging from 10% to 30% [1]. Sclerotherapy is a minimally invasive treatment option that involves the injection of a surfactant solution into insufficient veins [2,3]. Most commonly, the surfactant is mixed with room air to create sclerosing foams, in order to reduce the sclerosant’s consumption and deactivation by blood components [4,5,6]. Upon injection, sclerosing foams displace blood and maximise contact between the surfactant and inner walls of the target vessel. The surfactant reduces the surface tension of its environment, lysing the vascular endothelium, and resulting in vascular sclerosis [7].

Numerous previous studies have focused on bench-top physical characterisation of sclerosing foams including determination of bubble size distribution and foam’s half-life, with respect to factors such as surfactant concentration and liquid-to-gas ratio [8,9,10,11,12,13]. However, there is very limited research that rigorously and systematically studies the flow behaviour of sclerosing foams [14]. Furthermore, bench-top measurements of foam half-life and bubble size distribution do not reflect the behaviour of flowing foams upon administration, such as their capability in displacing blood inside a vein. Thus, it is essential to define and characterise clinically relevant parameters to correlate foam physical properties with clinical efficacy.

A limited number of studies have investigated the correlation of a sclerosing foam stability with the biological outcomes of sclerotherapy using endothelial cells in vitro [6,15,16,17] and on umbilical vein segments ex vivo [15]. In contrast, the effect of sclerosing foam rheological properties on therapeutic outcomes has not been evaluated yet. Previous studies have defined a clinically relevant parameter, such as the dwell time in vessels, to characterise the foam-vessel contact time [11,18,19]. Here, we aim to take a step further and measure the blood-displacement capability of sclerosing foams. Displacement flow of two miscible fluids (i.e., miscible displacement flow) is governed by buoyancy and is categorised into density-stable and density-unstable cases, based on the relative densities of the displacing and the displaced fluid. In the density-stable case, a less dense displacing fluid is used to displace a denser fluid. In contrast, in the density-unstable case, the displacing fluid is denser compared to the displaced fluid [20,21,22,23]. Density-unstable displacement flow of two miscible fluids is a well-studied field, both experimentally [20,21,22,23,24,25,26,27,28] and mathematically [29,30], with applications in removal of yield-stress fluids from obstructed pipes [24,25]. It has also been demonstrated that density-unstable displacement flow depends on the viscosity ratio of the two fluids and the inclination of the conduit [24,25]. The displacement of blood by sclerosing foams however fits within the density-stable displacement flow, given the greater density (ρ) of the displaced fluid (i.e., blood, $\rho_{blood}$ in the range 1043–1060 kg·m^−3^) [31] compared to that of the displacing fluid (i.e., sclerosing foams, $\rho_{foams}$ in the range 167–250 kg·m^−3^, estimated as weighted-average density of water and air for foam liquid-to-gas ratios of 1:3, 1:4 and 1:5) [10]. Furthermore, all previous studies on density-stable flows investigate the pumping of the displacing fluid at the inlet of a pipe, while during sclerotherapy the displacing fluid is injected via a needle or catheter.

Density-stable displacement has been shown to be generally more efficient compared to the density-unstable case [32], although the effect of viscosity contrast between fluids has not been studied. Etrati et al. demonstrated that for a downwards density-stable flow along an inclined pipe, greater viscosity of the displacing fluid increases displacement efficiency [22]. Therefore, it can be hypothesised that more viscous foam formulations would yield greater displacement efficacy. Previous studies characterised displacement efficacy by correlating interfacial velocity against average velocity of the displacing fluid. Sclerotherapy however is different from these previous investigations with respect to the modality of injection of the displacing fluid. Therefore, a different method is required in order to characterise displacement efficacy of sclerosing foams. Here, we attempt at correlating the viscosity of clinically used sclerosing foams with the displaced mass of a blood surrogate solution. The purpose of the blood surrogate is to mimic the rheological properties of blood. Additionally, red blood cells have been shown to consume sclerosants, which can change foam-blood interfacial interaction and in turn affect displacement flow of blood [4,5,6]. Therefore, in order to specifically evaluate the effect of foam’s viscosity on blood displacement efficacy, a blood surrogate solution was used instead of whole blood. The approach developed in this study will demonstrate the ability of different sclerosing foams to displace blood away from the treated vein segment. Thus, a clear link may be constructed between rheology of sclerosing foams and therapeutic outcomes of sclerotherapy. Findings from this study may inform the development of more effective therapeutic foam formulations.

## 2. Materials and Methods

In the present study, a pipe viscometer was employed to obtain a rheological model of physician-compounded-foams (PCFs), while a custom-designed method was employed to quantify the effectiveness of foams at displacing a blood surrogate. Data processing was conducted using Python to obtain apparent viscosity profiles and plot blood displacement efficacy against apparent viscosity of the foam.

### 2.1. Physician Compounded Foam (PCF) Formulation

The most commonly employed sclerosing foam formulations involve the use of either sodium tetradecyl sulfate (STS) or polidocanol (POL) surfactant solutions at volumetric concentrations of 0.5–3%. The liquid-to-gas ratio in a foam may define its stability and rheology [14,33]. In this work, sclerosing foams were formulated using volumetric liquid-to-gas (L:G) ratios of 1:3, 1:4 and 1:5 to evaluate the effect of foam dryness. Since previous studies demonstrated the almost identical therapeutic efficacy of 1% and 3% POL formulations [34,35], here we used a 1% *v*/*v* POL solution in phosphate-buffered saline (Croda, Goole, UK). Sclerosing foams were formulated using silicon-free Norm-Ject^TM^ syringes (BD Biosciences, Franklin Lakes, NJ, USA) and via two different clinically relevant PCF production methods, i.e., the double syringe system (DSS) and Tessari (TSS). In both methods, a 5 mL syringe filled with POL liquid solution was pushed in and out of a 10 mL syringe filled with room air, and the process was repeated 10 times. Volumes of surfactant solution and room air were adjusted to obtain 10 mL of foam with the desired L:G ratios (1:3, 1:4 and 1:5). In the DSS method, the syringes were connected via a Combidyn connector (B. Braun Melsungen, Germany) [18], while the Tessari method utilised a three-way stopcock (Cole-Parmer, UK) with an offset of 30° [36] (Figure 1).

### 2.2. Measuring Foam Apparent Viscosity Using Pipe Viscometry

The rheological behaviour of sclerosing foams was determined using pipe viscometry, as described in detail in our previous study [14]. A summary of the experimental parameters used in this work is included in Table 1. Pipe viscometry was conducted using a 15 cm long segment of polytetrafluoroethylene (PTFE) tubing (Cole-Parmer, UK) with inner diameter (ID) of 4.48 mm (see Figure 2a) to mimic the inner diameter of the healthy (4.41 ± 0.96 mm) and pathological (6.39 ± 2.21 mm) great saphenous veins [37]. PTFE was chosen as the constitutive material, as it represents a good compromise between optical transparency and a biologically relevant surface wettability (contact angle of water = 106.2° for PTFE [38] and ~90° for arterial endothelium [39]). After production, sclerosing foams were injected into the pipe viscometer using a syringe pump (PHD 2000 Programmable, Harvard Apparatus, Cambourne, UK). To avoid injecting the continuously drained liquid inside the syringe or exogenous air pockets, only 8 mL of the produced foam were delivered into the pipe. Two MPS2 pressure transducers (measurement limit 1000 ± 3 mbar, Elveflow, Paris, France) were connected to either side of the PTFE tube that provided pressure differential values used to model sclerosing foam rheology. Rheograms were constructed by plotting wall shear stress ($\tau_{w}$) against observed wall shear rate (${\dot{\gamma }}_{obs}$), calculated using the following equations:(1)$${\dot{\gamma }}_{obs}=\frac{32Q}{\pi d^{3}}$$
(2)$$\tau_{w}=\frac{d\Delta P}{4L}$$
where Q is the imposed volumetric flowrate, d is tube inner diameter (ID), L is tube length, and ΔP is the mean of time-averaged pressure differentials obtained during each repetition of pipe viscometry. Rheological data were fitted into the power law model (described in Equation (3)). Data were fitted into the linearised power-law equation (Equation (4)) in Prism 9 (Graphpad Software Inc., San Diego, CA, USA) to yield the fluid consistency index (K) and the fluid flow index (n).
(3)$$\tau_{w}=K{\dot{\gamma }}_{obs}{}^{n}$$
(4)$$\ln(\tau_{w})=n\cdot \ln{\dot{\gamma }}_{obs}+\ln K$$

Foam’s apparent viscosity ($\mu_{app}$) was then calculated using Equation (5), and subsequently used as the *x*-axis in the displacement flow curves (Equations (4) and (5)):(5)$$\mu_{app}=K{\dot{\gamma }}_{obs}{}^{n-1}$$

A detailed procedure for the data processing steps of the pipe viscometry experiments is also included in the Appendix A.

### 2.3. Selection of a Blood Surrogate Fluid

For the capability of a defined volume of foam to displace blood (or a blood substitute) to be evaluated, a rheological replica of blood must be produced. Previously, researchers have used 30% *v*/*v* glycerol solutions to mimic blood; however, blood is a non-Newtonian shear-thinning fluid [40,41] while glycerol solutions are Newtonian fluids [42,43]. A better substitute therefore may be carboxymethyl cellulose (CMC) that has been demonstrated to exhibit a shear-thinning behaviour [44,45]. Benchabane et al. have characterised the rheology of a variety of CMC solutions using the Cross model (see Equation (6)). This model can thus be used to calculate apparent viscosity of the blood surrogate; where $\mu_{0}$ and $\mu_{\infty }$ represent viscosity values at zero and infinite shear rates ($\dot{\gamma }$), while $\lambda_{c}$ and m represent a time constant and a dimensionless rate constant, respectively [44].
(6)$$\mu_{app}=\frac{\mu_{0}}{1+{\left(\lambda_{c}\cdot \dot{\gamma }\right)}^{1-m}}$$

Benchabane et al. and Shibeshi et al. have characterised the Cross model of CMC (0.65–0.85 degree of substitution, Sigma-Aldrich, St. Louis, MO, USA) and blood with great accuracy. Their findings were used to identify the most suitable CMC concentration for rheological replication of blood.

Prior to the blood displacement experiments, apparent viscosity of different CMC concentrations was compared to that of blood, to find the CMC concentration that provided the closest rheological replica to blood. The most biomimetic concentration of CMC was selected to be 0.2% *w*/*v* (see Appendix A). Thus, a 0.2% *w*/*v* CMC solution was produced in a 1 L volumetric flask, by mixing the required amount of CMC in distilled water (at 50 °C and 600 rpm, for 12 h). The density of the CMC solution was calculated by taking the mass of the CMC solution as the difference between the mass of the flask when empty and filled, and dividing it by flask volume (1 L).

### 2.4. Displacement Flow of Sclerosing Foams and a Blood Surrogate

Following pipe viscometry experiments and the determination of foams’ rheograms, different PCF formulations were tested for displacement flow efficacy inside the 4.48 mm PTFE tube. The experimental apparatus used for blood displacement experiments is illustrated in Figure 2b.

For each test, the PTFE tube was first primed with the 0.2% *w*/*v* CMC solution. Subsequently, 7 mL of foam was injected using a programmable syringe pump (AL-1000 Aladdin Single-Syringe Infusion Pump, WPI) while collecting the displaced CMC that left the tube. To model a wide spectrum of injection rates that may be used by clinicians during sclerotherapy, flowrates ranging between 4–36 mL·min^−1^ were employed. The mass of the displaced CMC was measured (using a Fisherbrand™ MH-124 analytical balance) and converted to volume using density of the CMC solution (984.6 ± 0.3 kg·m^−3^). The greatest source of error during these experiments was likely to be the manual nature of the CMC collection process, and particularly the selection of the collection end-point once foam injection was complete. In order to minimise this error, the length of the PTFE tube was maximised (60 cm) and 10 independent measurements were taken.

Given the shear-thinning nature of sclerosing foams (see Figure 3), different injection flowrates yield different apparent foam viscosities. Thus, the volume of displaced CMC was plotted against foam apparent viscosity in Figure 4a, calculated using Equation (5) from the rheological data obtained via pipe viscometry. The CMC displacement data were fitted into an asymptotic regression model (Equation (7)) to obtain the trendlines in Figure 4a.
(7)$$V_{CMC}=V_{m}-\left(V_{m}-V_{0}\right)e^{-\kappa \mu_{App}}$$
where $V_{m}$ represents the asymptote (equal to 7 mL in this case), $V_{0}$ is the *y*-intercept, and κ is a constant with units of (Pa·s)^−1^. Asymptotic parameters ($V_{0}$ and κ) were calculated through curve fitting at $V_{m}$ = 7 mL and were plotted against foam’s gas fraction ($\phi_{g}$) (see Figure 4b,c). Linear regression was conducted on $V_{0}$ and κ against $\phi_{g}$, to obtain empirical relationships between the asymptotic parameters and $\phi_{g}.$

In addition, the displaced volume of CMC was normalised with respect to the volume of gas present in the foam ($V_{g}$), resulting in a dimensionless parameter $\left(\frac{V_{CMC}}{V_{g}}\right)$ that was plotted against $\mu_{App}$ (Figure 5a). Trendlines were obtained by fitting the data in Equation (8). Following curve fitting, asymptotic parameters ($V_{0}/V_{g}$ and κ) were plotted against $\phi_{g}$ and linear regression was employed (see Figure 5b,c).
(8)$$\frac{V_{CMC}}{V_{g}}=V_{m}\phi_{g}-\left(V_{m}\phi_{g}-\frac{V_{0}}{V_{g}}\right)e^{-\kappa \mu_{App}}$$

Curve fitting of the asymptotic regression model was carried out in Python using the scipy.opimize.curve_fit function, while linear regression of the asymptotic parameters was conducted using the scipy.stats.pearsonr function. The pearsonr function returns the Pearson coefficient of correlation (R) while also testing a hypothesis for non-correlation, which calculates the probability that an uncorrelated set of data would produce the examined dataset. Following regression analysis, 2-way analysis of variance (ANOVA) was employed on the normalised displacement data to evaluate the extent of interaction between the independent parameters (e.g., foam injection flowrate, formulation technique, and L:G ratio) and the normalised data.

## 3. Results and Discussion

### 3.1. Characterisation of Foams’ Apparent Viscosity against Different Formulations

Processing of pressure differentials obtained via pipe viscometry using a 4.48 mm ID PTFE conduit returned the rheograms illustrated in Figure 3a. In this work, employing flowrates of 4–36 mL·min^−1^ resulted in observed wall shear rate (${\dot{\gamma }}_{obs}$) ranging ~ 7.55–68.0 s^−1^ and calculated wall shear stress ($\tau_{w}$) in the range ~ 19.17–60.96 Pa. Subsequently, data were fitted using a simple linear regression model into the linearised power-law model (Equation (4)) to obtain fluid consistency indices (8.142 < K < 12.49) and fluid flow indices (0.282 < n < 0.445) for each of the six foam formulations (three different L:G ratios and two different PCF formulation techniques). Finally, foam apparent viscosities ($\mu_{app}$) were calculated using Equation (5) for the range of ${\dot{\gamma }}_{obs}$ using the power-law indices, and were found to be in the range ~ 0.568 to 3.40 Pa·s (see Figure 3b). Results of linear regression of the linearised power law model (Equation (4)), as well as the resulting power law indices, are summarised in the Appendix A.

Slopes of the rheograms in Figure 3a decrease with increasing shear rate, indicating a shear-thinning flow behaviour amongst sclerosing foams. This is in agreement with previous studies on aqueous foams [46,47,48], including on sclerosing foams [13,14]. In addition, Figure 3b suggests that increasing the foam’s gas fraction (i.e., corresponding to lower L:G ratios) leads to an increase in foam apparent viscosity, which confirms previous findings [49]. This trend was consistent among both formulation techniques, although the rheograms corresponding to DSS foams cover a broader range of $\tau_{w}$ (19.17–60.96 Pa) compared to that of Tessari foams (22.96–57.48 Pa). Therefore, among 1:5 foams, DSS is expected to yield more viscous foams while for 1:3 foams it is Tessari foams that are predicted to be more viscous. This prediction is confirmed in Figure 3b. Importantly, the rheological model was used to calculate foam’s apparent viscosities at the imposed flowrates used in the displacement experiments, to enable a correlation between foam viscosity and blood surrogate displacement data.

### 3.2. Quantification of Sclerosing Foams’ Efficacy at Displacing a Blood Surrogate

Figure 4a shows the volume of displaced CMC solution (0.2% *w*/*v*) plotted against apparent viscosity of foam (calculated using Equation (5)). For all foam formulations, a positive asymptotic plateau relationship exists between $\mu_{app}$ and $V_{CMC}$ assuming that the maximum possible volume of displaced CMC is the same as the volume of injected foam (i.e., $V_{m}=V_{foam}=$ 7 mL). This relationship may be a result of the shear-thinning properties of sclerosing foams and as a result, decreasing flowrate would increase $\mu_{app}$ and lead to an increase in displacement efficacy, confirming our initial hypothesis. On the other hand, results in Figure 4a show that given a constant flowrate, increasing foam gas fraction causes a decrease in displacement efficacy. This finding is in contrast with what was expected. As shown in Section 3.1 above, drier foams exhibit greater viscosities and are expected to be more effective at displacing blood. However, the displacement experiments show that at a constant flowrate, drier foams are not as consequential as wet foams. While stating our hypothesis (i.e., greater viscosity → greater displacement efficacy), we assumed that wet and dry foams are only different with respect to viscosity. This is in fact not true as drier foams are more compressible than wet foams due to their greater gas content. So, at any given flowrate, drier foams may be compressed more, causing their volume to decrease which can explain their lower efficacy in displacing a blood surrogate solution in these experiments. Finally, different PCF formulation techniques showed no correlation among 1:4 and 1:5 foams, while for 1:3 foams the data trend towards an increased blood displacement efficacy for DSS foams (although this is not statistically significant). In summary, viscosity positively correlates with blood displacement efficacy of sclerosing foams of identical compressibility. In foams of varying compressibility, viscosity is no longer the sole indicator of foam displacement efficacy. Among such formulations, under a controlled injection rate, the least compressible foam formulation is expected to be most effective at displacing blood during injection. Finally, choice of PCF formulation technique (DSS or Tessari) offers no advantage with respect to blood displacement efficacy.

Upon fitting the displacement data into the asymptotic regression model (Equation (7)), $V_{0}$ and κ values were calculated and subsequently plotted against $\phi_{g}$ (Figure 4b,c). There is a strong correlation between κ and $\phi_{g}$ ($R^{2}$ = 0.9891, *p*-value = 0.0665), whereas $V_{0}$ shows a weaker linear relationship with $\phi_{g}$ ($R^{2}$ = 0.8843) that is in fact statistically insignificant (*p*-value = 0.2210). To improve the extent of statistical significance, the displacement data ($V_{CMC}$) were normalised with respect to volume of the constituent room air in sclerosing foams ($V_{g}$). The resulting normalised displacement plot (Figure 5a), alongside the corresponding linear regression of the asymptotic parameters ($V_{0}/V_{g}$ and κ), are illustrated in Figure 5b,c.

For both the original and normalised displacement data, the Pearson linear regression results of the asymptotic parameters against $\phi_{g}$ are summarised in Table 2, while values of the asymptotic parameters are included in the Appendix A. In both cases, linear regression results of κ against $\phi_{g}$, as well as the resultant values of κ, are identical and statistically significant with at least 90% confidence (*p*-value < 0.10). On the other hand, prior to normalisation, the $V_{0}-\phi_{g}$ data show no linear correlation (*p*-value = 0.2210). Nevertheless, after normalisation, the $V_{0}/V_{g}-\phi_{g}$ data also show a strong linear correlation with 95% confidence (*p*-value = 0.0288). Thus, assuming an asymptotic relationship between gas-normalised displaced volume of blood surrogate ($V_{0}/V_{g}$) and apparent viscosity of the sclerosing foams ($\mu_{app}$) via Equation (8), it can be inferred with at least 90% confidence that a linear relationship exists between gas-normalised asymptotic parameters ($V_{0}/V_{g}$ and κ) and $\phi_{g}$ that can be described through the following linear relationships:(9)$$\frac{V_{0}}{V_{g}}=\left(5.9149\pm 0.2987\right)\phi_{g}-\left(0.0887\pm 0.0551\right)$$
(10)$$\kappa =\left(5.8871\pm 1.0043\right)\phi_{g}-\left(0.2732\pm 0.1832\right)$$

Given a specific sclerosing foam gas fraction ($\phi_{g}$), $\frac{V_{0}}{V_{g}}$ and κ can be calculated using Equations (9) and (10). By inputting $\frac{V_{0}}{V_{g}}$ and κ into Equation (8) $\left[\frac{V_{CMC}}{V_{g}}=V_{m}\phi_{g}-\left(V_{m}\phi_{g}-\frac{V_{0}}{V_{g}}\right)e^{-\kappa \mu_{App}}\right]$, the volume of displaced blood surrogate can be predicted given a certain volume of foam ($V_{m}$). Thus, this work provides an empirical framework that can predict the displacement efficacy of a given sclerosing foam formulation.

Following the successful empirical formulation of displacement flow, 2-way ANOVA was employed to characterise the level of statistical significance amongst the normalised CMC displacement against different foam injection flowrates, formulation techniques, and L:G ratios. Detailed 2-way ANOVA results are included in the Appendix A). Results show almost no statistical significance between data obtained using different types of manual formulation technique (DSS vs. Tessari) when comparing foams at constant flowrate and L:G ratio (see Appendix A). Conversely, L:G ratio appeared to influence foam displacement efficiency, with wetter foams yielding greater CMC displacement per volume of gas ($V_{CMC}/V_{g}$ in range 1.62–1.75 and 1.01–1.14 for 1:3 and 1:5 foams, *p*-value < 0.001). This may be due to the greater compressibility of drier foams, which may explain their lower efficiency at displacing CMC during injection. In a previous study, Carugo and co-authors showed that 1:7 sclerosing foams exhibit greater drainage stability when compared to 1:4 foams [18]. However, in the present work, drier formulations were less efficient at displacing blood. Whether foam stability can predict displacement efficacy or not, is a question that remains unanswered. Nevertheless, it can be hypothesised that a larger gas fraction would result in a more stable foam that is also more compressible, which can lead to a lower displacement efficacy. Moreover, it is noteworthy to state that the sole focus of this study is on displacement flow upon injection, although further blood displacement may follow after injection due to the expansion of the compressed foam plug in absence of the external force of injection. A major difference between this work and that of Carugo et al. is tube orientation. While in this work the tube is oriented horizontally, the work of Carugo et al. involves injection of foam into inclined tubes. As a result, buoyancy contributes to the upward flow of foam in their study. On the other hand, flow of sclerosing foam presented in this work is independent of buoyancy, thus direct comparison with the work of Carugo et al. is not possible. Concerning foam injection flowrate, it was found that reducing the flowrate resulted in greater CMC displacement per volume of gas (*p*-value < 0.001), which also corresponded to greater foam viscosity. This implies that the effect of flowrate on apparent viscosity and CMC displacement dominates the effect of foam dryness and, as described above, this can be attributed to the greater compressibility of drier foam formulations.

To the best of the authors’ knowledge, this is the first study reporting on the displacement flow of sclerosing foams. The methods employed herein allow for characterisation of the blood displacement capabilities of sclerosing foams with respect to their viscosity, enabling the identification of the most cohesive foam formulation with an optimal blood displacement efficacy. Additionally, it was possible to correlate the asymptotic trends of CMC displacement per volume of constituent gas, with the gas fraction of sclerosing foams ($\phi_{g}$). This will allow users (i.e., clinicians) to not only predict the PCF formulation that is the most efficient at displacing blood, but also to obtain a quantitative measure of how different PCF formulations behave in comparison to one another.

### 3.3. Prediction of Clinical Performance

In order to predict clinical performance of sclerosing foams, their displacement efficacy on a blood surrogate solution (0.2% *w*/*v* CMC) was characterised as a clinically relevant parameter. Results show that lower foam injection flowrates would be more proficient at displacing blood. Coupled with the shear-thinning nature of sclerosing foams, it follows that lower flowrates increase foam viscosity, which in turn may be responsible for the observed increase in displacement efficiency. Given the shear-thinning nature of the blood surrogate, it should be noted that its viscosity would also decrease with decreasing its displacement rate.

On the other hand, drier foams were shown to be less effective at blood displacement despite their greater viscosity. This implies that the effect of flowrate on viscosity and blood displacement dominates the effect of gas fraction. Based on these results, it follows that—for any given PCF production technique or L:G ratio—lower injection rates would yield better blood displacement efficacy. This study predicts that PCF formulations of 1:3 L:G ratio would be the most effective at displacing blood at 4 mL·min^−1^, while formulation technique plays an insignificant role for manually produced foams.

Although here we show that more compressible PCFs are predicted to yield lower blood displacement efficacy during foam injection, it is still possible for more compressible foams to yield better therapeutic outcomes. Further studies are required to verify that greater blood displacement efficacy leads to better biological outcomes. For instance, a very dry formulation such as Varithena^TM^ (with a L:G ratio of 1:7) has been tested on ex vivo umbilical vein segments and was shown to be more effective at lysing endothelial cells to other PCF formulations of lower gas fractions [15]. In this previous work, Bottaro and co-authors performed a biological evaluation of room air PCFs and Varithena^TM^ (which is formulated using a mixture of oxygen and carbon dioxide gases), whereas the present work involves physical characterisation of room air PCFs. Future studies are needed to verify the therapeutic desirability of efficient blood displacement for foams of other formulation techniques and gas contents. Furthermore, our bench-top model only uses PTFE tubing and a blood surrogate solution. Therefore, the precision of the empirical framework developed here to predict blood displacement may need to be tested and evaluated using either ex vivo vein segments or biomimetic in vitro models (such as vein-on-a-chip), using whole blood.

## 4. Conclusions

While manually produced sclerosing foams (i.e., PCFs) have been employed as effective therapeutics to treat varicose veins, physical properties of different formulations have not been thoroughly correlated with clinically relevant parameters. Bottaro and co-authors [15] correlated formulation techniques (PCFs vs. Varithena^TM^) and needle diameter with sclerosing foam drainage kinetics and lytic activity. However, the extent of studies focusing on the rheological characterisation of sclerosing foams remains very limited. Furthermore, there is no previous study characterising the capability of sclerosing foams to displace blood upon injection into varicose veins. In this work, a therapeutically relevant rheometry apparatus—i.e., a pipe viscometer—was utilised to characterise the rheology of sclerosing foams using a power-law model. We also report the first case of using a carboxymethyl cellulose solution as a rheological replica of blood. Moreover, foam efficacy at displacing this blood surrogate medium was correlated with apparent viscosity. The volume of displaced blood surrogate obeyed an asymptotic model with foam apparent viscosity. In addition, the asymptotic parameters showed a strong linear relationship with foam gas fraction. Displacement flow results suggest that flowrate dominates the effect of L:G ratio on foam viscosity and displacement efficacy, for all PCFs investigated. Future studies may evaluate and verify the empirical asymptotic model of blood displacement on the displacement of whole blood by sclerosing foams, inside biomimetic conduits such as ex vivo vein segments or in vitro vein-on-a-chip models.

To the best of the authors’ knowledge, this work provides a novel approach to the correlation of sclerosing foam physics with clinically relevant parameters. Nevertheless, there are a few limitations worth noting. The manual handling of valves during displacement flow experiments could introduce measurement errors, and the process may thus be automated in the future to increase accuracy. Additionally, the interaction of sclerosing foams with venous endothelium was neglected in this work; future studies may explore displacement flow of sclerosing foams inside umbilical vein segments. This would be a challenge given that contact with sclerosing foams would alter integrity of the vascular structure after each injection. So, for consistency, a new vascular segment must be used for each injection, which would increase experimental costs and reduce throughput. Additionally, use of whole blood would potentially yield different displacement profiles, as blood directly consumes sclerosant molecules at affects the foam-blood interface. Finally, this study neglects the effect of the foam ageing phenomenon on the displacement efficacy of sclerosing foams. It is also important to note that this work solely focuses on PCFs that are formulated in-house by clinicians using room air. Foam formulations that use other gases or different formulation apparatuses (such as Varithena^TM^) that exhibit vastly different drainage kinetics, are expected to also exhibit a different rheological behaviour. This may therefore result in different displacement characteristics compared to physician compounded foams.

## Figures and Tables

**Figure 1 biomolecules-12-01725-f001:**
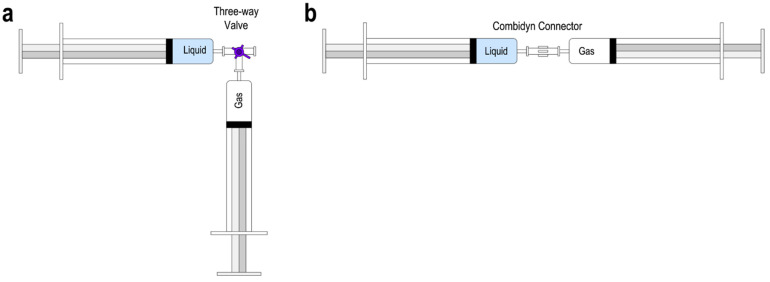
Methods of producing PCFs. 10 mL and 5 mL syringes are filled with the required amounts of room air and 1% *v*/*v* polidocanol solution, respectively. PCFs are formulated by pushing the liquid in and out of the syringes 10 times. (**a**) The Tessari (TSS) method. A three-way stopcock connects the syringes (the stopcock in this study is set at a 30° angle). (**b**) The double syringe system (DSS) method. Instead of a three-way stopcock, a Combidyn connector is used.

**Figure 2 biomolecules-12-01725-f002:**
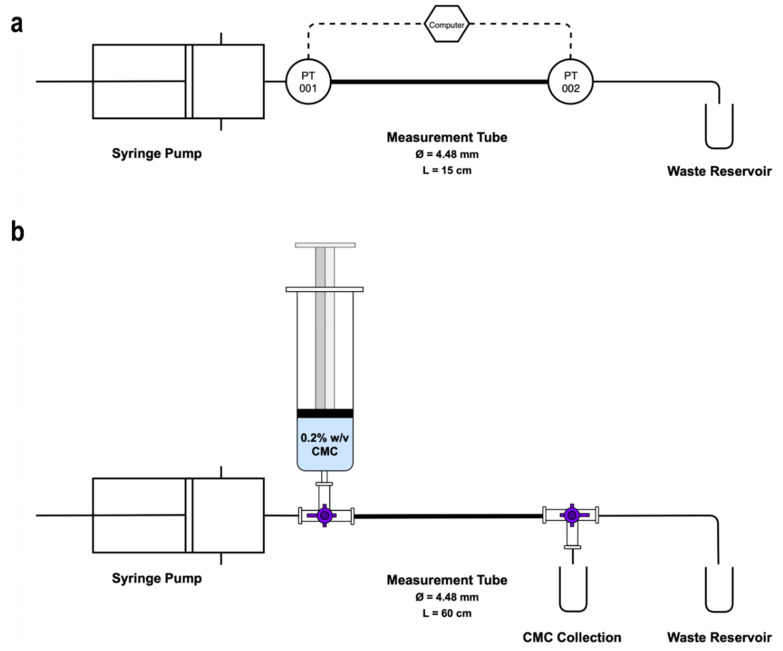
(**a**) Pipe viscometry apparatus. PCFs are injected into the measurement tube using a syringe pump. A computer interface is connected to the two pressure transducers (PT 001 and PT 002) that measure and record static pressure. (**b**) Blood displacement apparatus. Prior to each PCF injection, the measurement tube is purged with CMC three times and is subsequently primed with 0.2% *w*/*v* CMC solution using a 50 mL syringe. The inlet valve switches between CMC and foam injection while the outlet valve is set to CMC collection during foam injection and to waste during purge. Throughout foam injection, CMC is collected in a glass vial. Immediately at the end of injection, the second three-way valve is turned to stop CMC collection.

**Figure 3 biomolecules-12-01725-f003:**
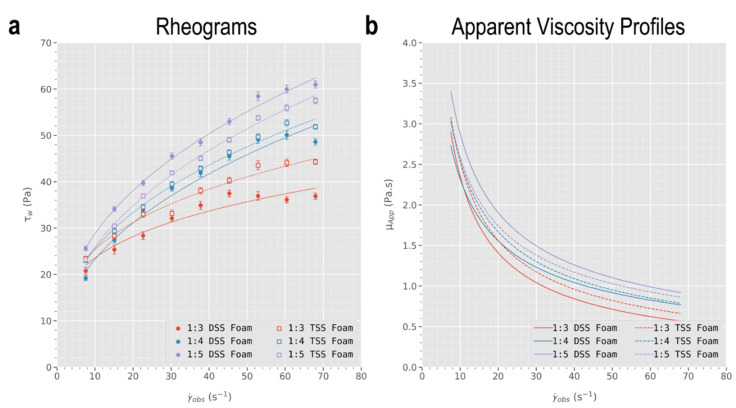
(**a**) Rheograms with non-linear power-law fits and (**b**) apparent viscosity profiles of DSS and TSS foams. The decreasing slopes of the rheograms and the declining trendlines of apparent viscosity indicate a shear-thinning behaviour of the foams. It is also evident that increasing gas fraction leads to an increase in apparent viscosity.

**Figure 4 biomolecules-12-01725-f004:**
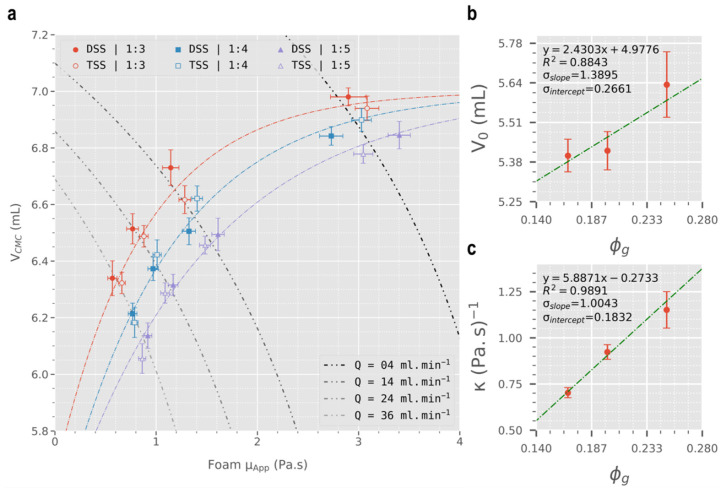
(**a**) Volume of CMC displaced versus foam apparent viscosity ($\mu_{app}$). $\mu_{app}$ was calculated using power-law indices obtained from the pipe viscometry experiments. Trendlines were obtained through curve fitting into an asymptotic regression model (see Equation (7)). Dotted lines represent flowrate isometric lines. (**b**) Linear regression of $V_{0}$ against $\phi_{g}$ (*p*-value = 0.2210). (**c**) Linear regression of κ against $\phi_{g}$ (*p*-value = 0.0665).

**Figure 5 biomolecules-12-01725-f005:**
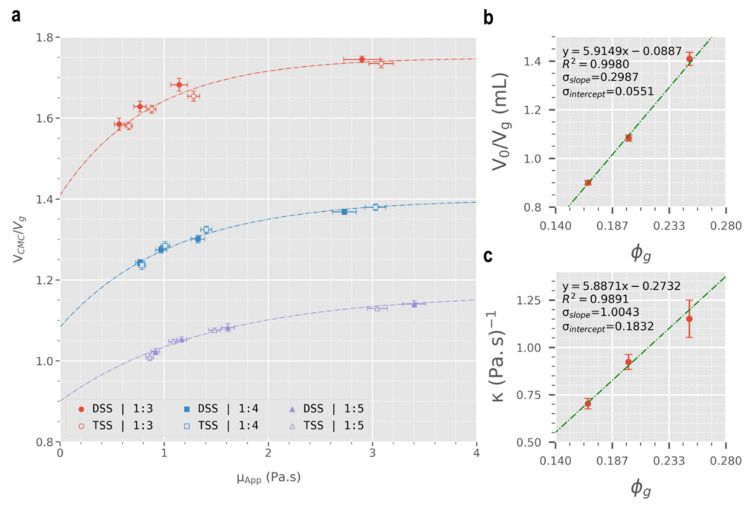
(**a**) CMC displacement per volume of foam-constituent-gas versus foam apparent viscosity ($\mu_{app}$). $\mu_{app}$ was calculated using power-law indices obtained from the pipe viscometry experiments. Trendlines were obtained through curve fitting into an asymptotic regression model (see Equation (8)). Dotted lines represent flowrate isometric lines. (**b**) Linear regression of $V_{0}/V_{g}$ against $\phi_{g}$ (*p*-value = 0.0288). (**c**) Linear regression of κ against $\phi_{g}$ (*p*-value = 0.0665).

**Table 1 biomolecules-12-01725-t001:** Summary of experimental parameters of pipe viscometry and blood displacement tests.

	Volume of Foam	PCF Formulations	Flowrates	Tube ID	Tube Length	Wall Shear Rates	Repeats
Pipe Viscometry	8 mL	DSS 1:3, 1:4, 1:5 TSS 1:3, 1:4, 1:5	4, 14, 24, 36 mL·min^−1^	4.48 mm	15 cm	7.55–67.97 s^−1^	3
Blood Displacement	7 mL				60 cm		10

**Table 2 biomolecules-12-01725-t002:** Results of linear regression of asymptotic parameters against $\phi_{g}$. The Pearson coefficient of determination (R^2^) and the *p*-value associated with hypothesis testing of non-correlation are also included.

Original Data						
	slope	$\sigma_{\mathrm{slope}}$	y-intercept	$\sigma_{y-\mathrm{intercept}}$	$R^{2}$	*p*-value
** $V_{0}-\phi_{g}$ **	2.4302	1.3895	4.9776	0.2661	0.8843	0.2210
** $\kappa -\phi_{g}$ **	5.8871	1.0043	−0.2733	0.1832	0.9891	0.0665
**Normalised Data**						
	slope	$\sigma_{\mathrm{slope}}$	y-intercept	$\sigma_{y-\mathrm{intercept}}$	$R^{2}$	*p*-value
** $V_{0}/V_{g}-\phi_{g}$ **	5.9149	0.2987	−0.0887	0.0551	0.9980	0.0288
** $\kappa -\phi_{g}$ **	5.8871	1.0043	−0.2732	0.1832	0.9891	0.0665

## Data Availability

The data presented in this study are available upon request.

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
