# Peer review of "Foam-in-Vein: Characterisation of Blood Displacement Efficacy of Liquid Sclerosing Foams"

_biomolecules, 2022, doi:10.3390/biom12121725_

Round 1

Reviewer 1 Report

The paper by Meghdadi et al. describes an in-vitro model to investigate flow behaviour of sclerosing foams. Flow behaviour of foams during treatment is not fully understood thus studies in this direction can help improve the insight into schlerotherapy treatment and lead to an optimised clinical outcome. The reviewer confirms that, to the best of his knowledge,  this is the first study dealing with displacement flow of sclerosing foams. In general, the paper is well written and useful. I would strongly recommend  its publication after few  minor revisions which I am summarizing below.

·         1) Introduction

Line 87: I would recommend to provide more explanations on why this hypothesis was considered in the first instance. 

·        2) Materials and Methods

Line 135: is shear rate in Equation 1 calculated at the wall?

Line 139: clarify that tube ID= tube inner diameter

·        3) Results and discussion

Line 272: I would recommend specifying that, since both blood replica and foams are shear thinning fluids, decreasing flow rate would lead to an increase of apparent viscosity in both fluids.

Figure 5 b and c: resolution is a bit low in these figures

Line 347: I would recommend to expand the comparison with previous results shown in Carugo et al.. I would suggest to list all possible differences in the experiments which may have played a role on the different reported efficacy of dry/wet foams in displacing blood.

Author Response

Comment 1 Line 87: I would recommend to provide more explanations on why this hypothesis was considered in the first instance.

Response 1: The text in this section has been edited to more clearly describe the formulated hypothesis.

Comment 2 Line 135: is shear rate in Equation 1 calculated at the wall?

Response 2: We thank the reviewer for pointing this out. This has now been clarified in the revised manuscript, where we have specified that Eq. 1 refers to wall shear rate.

Comment 3 Line 139: clarify that tube ID= tube inner diameter.

Response 3: We now have clarified that ‘ID’ corresponds to ‘inner diameter’.

Comment 4 Line 272: I would recommend specifying that, since both blood replica and foams are shear thinning fluids, decreasing flow rate would lead to an increase of apparent viscosity in both fluids.

Response 4: We thank the reviewer for their comment. We have now indicated that the apparent viscosity of both fluids would increase with decreasing the flow rate.

Comment 5 Figure 5 b and c: resolution is a bit low in these figures

Response 5: We agree with the reviewer, and have now edited both Figures 4 and 5 to improve their resolution.

Comment 6 Line 347: I would recommend to expand the comparison with previous results shown in Carugo et al.. I would suggest to list all possible differences in the experiments which may have played a role on the different reported efficacy of dry/wet foams in displacing blood.

Response 6: We agree with the reviewer that a more detailed comparison of the experimental conditions between studies could have been included. We have now expanded our discussion of this comparison in the revised manuscript.

Reviewer 2 Report

The authors reported the engineering of a novel device to model sclerosing foams rheology upon injection. This allowed for the generation of a unique platform for the testing and characterisation of sclerosing foams. Abstract and Introduction are clear. Aims and objectives are clearly stated and well explained. However, from line 115, references are missing with a constant message (Error! Reference source not found) visible at the end of the text. We invite the authors to amend for this issue throughout the entire manuscript. Moreover, the absence of a control is clear. We invite the author to revise the data here presented including a positive control (commercial formulation) which would greatly improve the comparability of the data and the impact of the study. Altogether, results are clearly presented, and the manuscript is well-casted. We recommend this manuscript for minor revisions.

Comments:

Introduction

Line 75: please include most typical pipes diameters, lengths, material used, justifying the use of the authors’ setup.

Line 86: please, clearly report a justification for the use of blood surrogate solution instead of human blood, describing potential differences in rheological behaviour.

Results and Discussion

We invite the authors to produce data on the CMC displacement in comparison to a control group. Data related to a clinically-used foam will greatly improve the impact of the study. This will clearly confirm the functionality of the proposed methodology elating this new model.

Line 294 and 310

Panels in Figure 4 and Figure 5 can be combined. We believe that the panels reporting the linear regressions can be ideally moved to the supplementary data within the manuscript.

Line 320-321

Please, do clarify this statement. The reader might find this not clear.

Line 361

We invite the authors to clarify the statement in line 361. Please, do compare this study with work produced by the authors in previous publications.

Supplementary information

Please, revise the citation of supplementary data within the manuscript main text. Most of the figures here reported are not cited within the main body.

Figure S2

We suggest the author for minor change related to the reduction of figures in the manuscript. Figure S2 and Figure 2 can be combined in a single figure presented as Figure 2.

Figure S4

It is not clear if the blood here used is human or not. Please, do provide further explanation. 

Author Response

Comment 1 Line 75: Please include most typical pipes diameters, lengths, material used, justifying the use of the authors’ setup.

Response 1: We agree with the reviewer that further information could be included to support some of the methodological choices made. A justification for the pipe diameter and material used has now been provided in the revised manuscript. As this is a displacement flow study, the length of the pipe is a less critical factor. In fact, the greater the length of the pipe, the more accurate would be the quantification of collected displaced fluid. This is because the blood substitute solution is displaced out of the tube in discrete droplets. The size of a droplet is the minimum amount of fluid that is displaced. On the other hand, the displacing fluid (i.e., foam) does not enter the tube in discrete amounts. Therefore, at times at the end of foam injection, a droplet of blood substitute would hang from the tube outlet, which is equal to the error in the measurement. The size of this error (relative to the total amount of displaced blood substitute) decreases with increasing tube length, because a longer tube can hold more fluid within its volume.

Comment 2 Line 86:  Please, clearly report a justification for the use of blood surrogate solution instead of human blood, describing potential differences in rheological behaviour.

Response 2: We thank the reviewer for pointing this out. The manuscript has now been revised to include a justification for the use of a blood surrogate.

Comment 3 Results and Discussion: We invite the authors to produce data on the CMC displacement in comparison to a control group. Data related to a clinically-used foam will greatly improve the impact of the study. This will clearly confirm the functionality of the proposed methodology elating this new model.

Response 3: Foams investigated in this study are clinically used. We have now added a clarification in the revised manuscript. Furthermore, section 2.1 begins with providing this information and then describing the choice of formulation in this paper, which is aligned with clinical formulations.

Comment 4 Line 294 and 310: Panels in Figure 4 and Figure 5 can be combined. We believe that the panels reporting the linear regressions can be ideally moved to the supplementary data within the manuscript.

Response 4: The authors appreciate the reviewer’s comment on this point. However, as Reviewer 1 requested an increase in resolution, the authors believe that Figures 4 and 5 can remain in the manuscript body. The fact that a linear relationship is evident increases the impact of these findings. Such clear trends in data most often point to the rigour of the experiments, and we believe that these figures would further help illustrating the impact of the work.

Comment 5 Line 320-321: Please, do clarify this statement. The reader might find this not clear.

Response 5: We thank the reviewer for their suggestion. We have now clarified this statement.

Comment 6 Line 361: We invite the authors to clarify the statement in line 361. Please, do compare this study with work produced by the authors in previous publications.

Response 6: Following the reviewer’s comment, we have included a comparison with our earlier study in the revised manuscript.

Comment 7 Supplementary Information: Please, revise the citation of supplementary data within the manuscript main text. Most of the figures here reported are not cited within the main body.

Response 7: Although individual supplementary figures and data are not cited, supplementary sections are all cited within the text. The authors avoided over citation of supplementary figures to minimise confusion among readers. When needed, readers are guided to specific supplementary sections throughout the manuscript so they can view this information if they wish to do so.

Comment 8 Figure S2: We suggest the author for minor change related to the reduction of figures in the manuscript. Figure S2 and Figure 2 can be combined in a single figure presented as Figure 2.

Response 8: We thank the reviewer for their recommendation. We however believe that Figure S2 provides supplementary information to the one already provided by Figure 2. We also feel that the overall number of figures (= 5) is appropriate for the manuscript. We have therefore kept Figure S2 in the Supplementary section of the revised manuscript.

Comment 9 Figure S4: It is not clear if the blood here used is human or not. Please, do provide further explanation. 

Response 9: We agree with the reviewer, and we have now clarified this within both the manuscript body and in the caption of Figure S4.